# Olaparib Combined with an ATR or Chk1 Inhibitor as a Treatment Strategy for Acquired Olaparib-Resistant *BRCA1* Mutant Ovarian Cells

**DOI:** 10.3390/diagnostics10020121

**Published:** 2020-02-22

**Authors:** Brian T. Burgess, Abigail M. Anderson, J. Robert McCorkle, Jianrong Wu, Frederick R. Ueland, Jill M. Kolesar

**Affiliations:** 1Division of Gynecologic Oncology, Department of OB/GYN, University of Kentucky, Whitney-Hendrickson Building, 800 Rose Street, Lexington, KY 40536, USA; brian.burgess@uky.edu (B.T.B.); fuela0@uky.edu (F.R.U.); 2Markey Cancer Center, University of Kentucky, 789 South Limestone Street, 526 Todd Building, Lexington, KY 40536, USA; aman239@uky.edu (A.M.A.); rob.mccorkle@uky.edu (J.R.M.); 3Biostatistics and Bioinformatics Shared Resource Facility, University of Kentucky, 800 Rose Street, Roach Building CC433, Lexington, KY 40536, USA; Jianrong.Wu@uky.edu; 4College of Pharmacy, University of Kentucky, 567 Todd Building, 789 South Limestone Street, Lexington, KY 40536, USA

**Keywords:** Olaparib, PARP, ATR, Chk1, resistance, BRCA1, ovarian cancer, inhibitor, UWB1.289, UWB1.289 + BRCA

## Abstract

Objective: Despite the promise of PARP inhibitors (PARPi) for treating *BRCA1/2* mutated ovarian cancer (OC), drug resistance invariably develops. We hypothesized rationale drug combinations, targeting key molecules in DNA repair pathways and the cell cycle may be synergistic and overcome acquired PARPi resistance. Methods: Drug sensitivity to PARPi alone and in combination with inhibitors of key DNA repair and cell cycle proteins, including ATR (VE-821), Chk1 (MK-8776), Wee1 (MK-1775), RAD51 (RI-1) was assessed in PARPi-sensitive (UWB1) and -resistant (UWB1-R) *gBRCA1* mutant OC cell lines using a cell proliferation assay. The Bliss synergy model was used to estimate the two-drug combination effect and pharmacologic synergy (Bliss score ≥ 0) or antagonistic (Bliss score ≥ 0) response of the PARPi in combination with the inhibitors. Results: IC_50_ for olaparib alone was 1.6 ± 0.9 µM compared to 3.4 ± 0.6 µM (*p* = 0.05) for UWB1 and UWB1-R cells, respectively. UWB1-R demonstrated increased sensitivity to ATRi (*p* = 0.04) compared to UWB1. Olaparib (0.3–1.25 µM) and ATRi (0.8–2.5 µM) were synergistic with Bliss scores of 17.2 ± 0.2, 11.9 ± 0.6 for UWB1 and UWB1-R cells, respectively. Olaparib (0.3–1.25 µM) and Chk1i(0.05–1.25 µM) were synergistic with Bliss scores of 8.3 ± 1.6, 5.7 ± 2.9 for UWB1 and UWB1-R cells, respectively. Conclusions: Combining an ATRi or Chk1i with olaparib is synergistic in both PARPi-sensitive and -resistant *BRCA1* mutated OC cell models, and are rationale combinations for further clinical development.

## 1. Introduction

Ovarian cancer (OC) is the deadliest gynecologic malignancy and the American Cancer Society estimates that there will be 22,530 new diagnosis and 13,980 deaths in 2019 [1]. In the last decade there has been a surge of genetic data and a new-found method for the treatment of OC with germline mutations in *BRCA1/2*. In the case of serous ovarian cancers, up to 25% of cases contain germline mutations with the most common being *BRCA1* or *BRCA2* mutations [2].

BRCA1/2 are multi-functioning tumor suppressor proteins that play a vital role in homologous recombination repair (HRR) of double-stranded DNA breaks (DSBs), cell cycle checkpoint activation, replication fork (RF) protection, and generating single-stranded DNA during repair after irradiation damage [3,4,5]. Defective HRR predisposes cancer cells to increased genomic instability and represents a unique vulnerability that can be exploited by anticancer therapy directed at complementary pathways. This concept led to the discovery that germline *BRCA1*- and *BRCA2*-mutated OCs are hypersensitive to poly(ADP-ribose) polymerase (PARP) enzyme inhibition—an enzyme that has an important role in DNA base excision repair [6,7]. As an anticancer treatment strategy, the combination of a *BRCA1/2* genetic mutation and a PARPi is often referred to as synthetic lethality [8]. Based on clinical trial success for patients with recurrent epithelial OC containing a germline *BRCA1* or *BRCA2* mutation, olaparib was approved in 2014 as a first-in-class PARP inhibitor (PARPi) for treatment of recurrent platinum sensitive OC with a demonstrated progression free survival benefit of 11.2 versus 4.3 months for maintenance therapy compared to placebo [9]. The overall survival benefit of 4.7 months, however, with olaparib maintenance monotherapy is less impactful and is believed to be in part related to development of an acquired PARPi resistance [10]. It is thought that tumor cells can develop clinical acquired PARPi resistance over time by two general mechanisms that includes either (a) returned functionality to the HRR pathway by perhaps a somatic restoration of the *BRCA1* mutation or evolution of an alternate *BRCA1/2*-independent HRR pathway or (b) mutations in PARP resulting in reduced PARP trapping [11]. Today, acquired PARPi resistance is a significant clinical barrier to improved long-term treatment success and even a potential cure in these patients.

HRR is a complex pathway that requires not only the efficient use of BRCA1 and BRCA2 proteins but a number of other related proteins including REV7, PTIP, RIF1 and RAD51 for example [12]. Cells with defective HRR pathway are shunted to non-homologous end joining or alternate end joining pathways to repair DSBs at the expense of increased genomic instability. Recently, a bypassing of the *BRCA1/2*-dependent role in HRR pathway was proposed in *BRCA1* mutant OC cells that involves an increased reliance on ataxia telangiectasia and Rad3 (ATR) protein for survival [13]. Interestingly, ATR is a large kinase which phosphorylates protein substrates and is regarded as principal direct effector of recruiting for DDR and cell cycle checkpoints [14]. ATR inhibition (ATRi) was shown to disrupt a restored *BRCA*-independent HRR and restored RF stability suggesting a compensatory role for ATR in olaparib-resistant *BRCA* mutant OC cells [13].

Lastly, in addition to a vital role HRR in actively dividing tumor cells, the activation of multiple mechanistically distinct cell checkpoint responses which halt the cell from progressing to the next cell cycle, facilitating DNA repair and promoting cell survival is equally a critical function of the cell [15]. Major cell cycle checkpoints include the G1/S and G2/M transition (also known as the G1 and G2 checkpoints) as well as the intra-S checkpoint which controls the rate of DNA synthesis. The G1 checkpoint is uniquely dependent on the p53 protein and in the case of serous OC has been found to be highly mutated (96% of cases) [16]. To that end, it has been argued that because of the high frequency mutation rate of the p53 protein in serous OC, tumor cells may be more dependent on a downstream functioning G2 checkpoint compared to their normal cell counterparts. As a result, the G2 checkpoint is an attractive anticancer target for *BRCA1/2* mutant OCs, which inherently have a deficiency in HRR [14,17].

The purpose of this study was to evaluate the ability of G2 checkpoint and HRR associated protein inhibitors to overcome PARPi resistance.

## 2. Materials and Methods

### 2.1. Cell Lines

UWB1.289 and UWB1.289 + BRCA cell lines were purchased from American Type Culture Collection (ATCC). Cells were maintained at 37 °C, 5% CO_2_ in a humidified incubator in 1:1 MEBM Bullet Kit and RPMI-1640 (Lonza) with 3% v/v FBS cell media. UWB1.289 + BRCA cells were maintained with aforementioned media plus 200 μg/mL G418 [18]. The UWB1.289 + BRCA cell line as purchased had previously been transfected with pcDNA3 plasmid carrying wild-type *BRCA1* gene as previously described [18]. Olaparib resistant cells were derived using UWB1.289 cells plated in a 12 well plate approximately 50% confluency and maintained in 1µM olaparib-containing media for 21 days. Olaparib-containing media was changed two times were week. After 21 days of continuous exposure, derived cells were passaged and stored in liquid nitrogen for future use.

### 2.2. Expression of BRCA1/2

Total RNA was extracted from UWB1.289 cells using the RNeasy Plus Universal Mini Kit (Qiagen). Maxima H Minus FirstStrand cDNA Synthesis Kit (ThermoFisher Scientific, Waltham, MA, USA) was used to prepare cDNA according to manufacturer’s instructions. Real-time PCR was run on each cDNA sample in triplicate per gene using TaqMan Gene Expression Assays (BRCA1: Hs01556193_m1, BRCA2: Hs00609073_m1) and a QuantStudio 7 Flex Real-Time PCR System (ThermoFisher Scientific, Waltham, MA, USA). Relative gene expression was calculated using the DDC_T_ method with TBP (Hs00427621_m1) as the endogenous reference gene [19].

### 2.3. Inhibitors

Cells were treated with the following inhibitors: PARP inhibitor (olaparib, Fisher Scientific, Waltham, MA, USA), a ChK1 inhibitor (MK-8776, SelleckChem, Houston, TX, USA), ATR inhibitor (VE-821, SelleckChem, Houston, TX, USA), Rad51 inhibitor (RI-1, SelleckChem, Houston, TX, USA), and a Wee1 inhibitor (MK-1775, SelleckChem, Houston, TX, USA). Aliquots of inhibitors were prepared from lyophilized product, dissolved in DMSO and diluted in cell media using serial dilutions to a final concentration DMSO concentration of 0.1% DMSO.

### 2.4. Cell Proliferation Assays

To determine the cytotoxicity response to the pharmacologic agents, cells were plated in opaque 96 well plates at 3000 cells/well with 100 µL media and incubated for approximately 24 h to allow for attachment to the plates. Drug-diluted media at varying target concentrations was prepared and the media exchanged in each well after 24 h. Cells were then incubated for 6 days. After 6 days, CellTiter-Glo 2.0 cell proliferation assay (Promega, Madison, WI, USA) was warmed to room temperature and 50 µL added to each well. An amount of 50 µL of CellTiter-Glo 2.0 was also added to the wells containing media only as well as wells with untreated cells (inhibitor-free media). Plates were then shaken for 2 min using an orbital shaker and allowed to equilibrate for a total of approximately 30 min. Using a Varioskan LUX (Thermo ScientificWaltham, MA, USA), the luminescent signal was measured and exported to a data file. Cell survival fraction was then determined by: (1)$$S=\left(\frac{\mathrm{ftreated}-\mathrm{fbackground}}{\mathrm{funtreated}-\mathrm{fbackground}}\right)$$
where ftreated represents luminescence of cells treated with cytotoxic drug, funtreated represents average luminesce of untreated cells, fbackground luminesce of cell media only. IC50 for each experiment was determined by fitting a dose-response model to cell survival fraction, S, as a function of increasing drug concentration using R studio software (version 3.6, Boston, MA, USA) [20] and the DRC package [21]. Multi-dimensional two drug synergy studies were also completed with the screened inhibitors. The drug combination effect was assessed using the Synergyfinder package [22] in R Studio to determine the Bliss independence model score [23]. In quantifying the response to drug interactions, the Bliss independence model works on the assumption that the combined drugs are acting independently (mutually non-exclusive) of one another. The reference model can be demonstrated by the common formula for Bliss independence:

E_c_ = E_a_ + E_b_ − E_a_(E_b_)
(2)
where E_c_ is the effect of the combination of two drugs (A and B) at given doses (a and b) [24]. The response to drug A at dose a is E_a_ and E_b_ indicates the effect of drug B at dose b. Deviation in observed effect greater than the expected effect determined from the reference model is considered a synergistic interaction; a negative deviation is considered an antagonistic combination. If the observed and expected effects are equivalent, the combination is deemed additive or noninteractive.

### 2.5. Long Term Drug Exposure Studies

Cells were plated in 12 well plates at approximately 50% confluency and drug-diluted media using olaparib (1 µM), VE-821 (0.8 µM), MK-8776 (0.5 µM), olaparib (1 µM) + VE-821 (0.8 µM), olaparib (1 µM) + MK-8776 (0.5 µM) or DMSO (0.2%, control) for 21 days continuously. Three replicates of each condition were prepared, and drug-diluted media was exchanged two times per week. After 21 days, viable cell counts were estimated using 0.4% trypan blue staining [25]. Cell survival fraction was estimated for each condition by normalizing to control cells cultured in drug free media.

### 2.6. Statistical Analysis

Statistical comparison between treatments was computed by students T-test (2-way comparisons) ANOVA (3-way comparisons) followed by Tukey’s Honest Significant Difference test using GraphPad Prism (version 5, San Diego, CA, USA) software. Significance of each test statistical test was assessed by α < 0.05, <0.01, and <0.001.

## 3. Results

To investigate how to overcome acquired PARPi resistance in *BRCA* mutant ovarian cancer cells, we used a well-established human high grade serous ovarian cancer cell line containing a frameshift germline *BRCA1* mutation (UWB1.289, hereafter referred to as UWB1). A second cell line previously derived from the UWB1 cell line was used as an experimental control and expressed a restored wild-type *BRCA1* function (UWB1.289 + BRCA, hereafter referred to as UWB1-WT) [18]. An olaparib resistant cell line was generated from UWB1 cells that were treated with sustained, physiologically relevant 1µM concentrations of olaparib-containing cell media for 21 days to produce an olaparib-resistant, *BRCA1* mutant ovarian cancer cell line named UWB1-R. Real-time PCR analysis of mRNA expression for BRCA1 was completed to assess the BRCA1 expression stability in the cell lines. From Figure 1A, the UWB1 and UWB1-R cell lines expressed relatively similar levels of mRNA of BRCA1, however the UWB1-WT cells expressed higher levels of BRCA1 mRNA expression, a finding consistent with a restored wild-type. To evaluate the cytotoxicity of olaparib to the parent cell line compared to the UWB1-R cells, we determined the IC_50_ using a cell viability assay. Dose–response curves are shown in Figure 1B for each of the cell lines. As expected, the *BRCA1* mutant cell line was the most sensitive to PARPi, with a mean IC_50_ in the UWB1 of 1.6 ± 0.9 µM, compared to the olaparib resistant, UWB1-R was 3.4 ± 0.6 µM and *BRCA* wild-type, UWB1-WT IC_50_ > 10 µM (IC_50_ not attainable). The difference in IC_50_ between UWB1 and UWB1-R cells represents a statistically significant (*p* = 0.05) increase of approximately 2.1X for UWB1-R, suggesting the development of PARPi resistance in the olaparib treated cell line (Figure 1C).

We hypothesized that the inhibitors of key DDR proteins and cell cycle checkpoint molecules would be synergistic with a PARP inhibitor for a *BRCA* mutant OC. We first assessed the single agent effects of ATRi (VE-821), a Chk1i (MK-8776), Wee1i (MK-1775), and RAD51i (RI-1) on cell proliferation, demonstrating UWB1-R cells were more sensitive to ATR inhibition than the parent UWB1 cell lines with a IC_50_ of 2.8 ± 0.77 µM for UWB1, compared an IC_50_ of 0.78 ± 0.04 µM in UWB1-R cells (*p* = 0.04) (Figure 2A and 2B). The IC_50_ of the UWB1-WT was 1.1 ± 0.08 µM and also trended lower compared to the UWB1 cells. For the Chk1i single agent treatment, the *BRCA* wild-type cells were the most sensitive. The IC_50_ of the UWB1-WT cells (0.39 ± 0.004 µM) were significantly lower compared to the UWB1 cells (*p* = 0.04). The IC_50_ for the UWB1-R cells (0.47 ± 0.02 µM) trended lower compared to UWB1 (0.69 ± 0.11 µM); however, the magnitude of difference was not statistically significant. For the Wee1i, the UWB1-WT cell line was significantly more sensitive (IC_50_ = 0.09 ± 0.005 µM) compared to UWB1 (0.11 ± 0.005 µM) and UWB1-R cells (IC_50_ = 0.13 ± 0.05 µM); however, in general, the magnitude of differences was small and all cell lines are deemed highly sensitive to Wee1i. No significant inhibition of cell proliferation was observed for the RAD51i up to a 1.5 µM for any of the cell lines, and therefore an IC_50_ estimate was not attained. Given the lower IC_50_ values observed for the ATR and Chk1 inhibitors with the UWB1-R cells compared to UWB1 we next hypothesized the combination of a PARP inhibitor with an ATRi or Chk1i would be synergistic and a potential clinical strategy to overcome PARPi resistance.

To evaluate drug combinations, a multi-dimensional two-drug synergy assay was completed using olaparib in combination with an ATRi, Chk1i, Wee1i, and RAD51i. The drug combination effect was assessed by the Bliss independence model [23]. If the combined effect is greater than what would be expected for each drug additively, then the response is classified as synergistic (and a Bliss score > 0), while antagonism is concluded when the combination produces less than the expected effect (Bliss score < 0). From Figure 3A, the olaparib plus ATRi (Olap + ATRi) was synergistic across all three cell lines with estimated Bliss scores of 17.2 ± 0.2, 11.9 ± 0.6 and 4.2 ± 1.5 for UWB1, UWB1-R and UWB1-WT, respectively. The Bliss scores of the *BRCA* mutant cell lines, UWB1 and UWB1-R, were significantly greater compared to UWB1-WT (*p* = 0.003). Combining olaparib with Chk1i (Olap + Chk1i) was also synergistic in the *BRCA1* mutant cell lines UWB1 (8.3 ± 1.6) and UWB1-R (5.7 ± 2.9) but antagonistic in the UWB1-WT line (−2.0 ± 1.2). The UWB1 Bliss score was significantly greater than UWB1-WT (*p* = 0.03). Olaparib + Wee1i (Bliss score UWB1: −0.66 ± 1.1, UWB1-R: −0.62 ± 0.25) and Olaparib + RAD51i (Bliss score UWB1: −0.29 ± 4.4, UWB1-R: −2.2 ± 1.09) were not synergistic in the *BRCA1* mutant cell lines, however synergy was noted for the UWB1-WT cell line with a Bliss score of 4.2 ± 2.1 and 6.4 ± 0.8 for Olap + Wee1i and Olap + RAD51i, respectively. None of the Bliss scores for Wee1i and RAD51i were statistically significant among the three cell lines.

As shown in Figure 3B, drug synergy for Olap + ATRi was demonstrated over a robust concentration range, with olaparib at concentrations ranging from 0.3–1.25 µM and the corresponding ATRi concentrations ranging from 0.8–2.5 µM. Cell inhibition was noted to be up to >90% with these concentrations ranges. Similarly, as shown in Figure 3C, drug synergy for Olap + Chk1i was found over a large concentration with olaparib 0.3–1.25 µM and the corresponding Chk1i 0.05–1.25 µM. The Olap + Chk1i also achieved up to >90% cell inhibition.

Focusing on the two drug combinations demonstrating robust drug synergy for UWB1 and UWB1-R cells, a long-term drug exposure study was conducted by treating the cells with single agent olaparib (1 µM), ATRi (0.8 µM) and Chk1i (0.5 µM) as well as the two-drug combination of Olap + ATRi (1 µM and 0.8 µM, respectively) and Olap + Chk1i (1 µM and 0.5 µM, respectively) for a continuous 21-day period. The individual drug concentrations for the olaparib, ATRi and Chk1i were selected based on the two drug synergistic concentration range determined previously (Figure 3B,C). From the results shown in Figure 4, UWB1 and UWB1-R had a cell viability of 6.4 ± 4.9% and 18.9 ± 1.7% respectively after 21 days of treatment with olaparib alone. Both UWB1 and UWB1-R cells were significantly (*p* = 0.0002) more sensitive to olaparib than UWB1-WT (103 ± 22% cell viability). Treatment with ATRi alone resulted in a similar cell viability trend of 9.6 ± 8.8%, 30.8 ± 5.3% and 74.2 ± 19% for the UWB1, UWB1-R and UWB1-WT cells. UWB1 and UWB1-R cells were significantly more sensitive to ATRi compared to UWB1-WT (*p* = 0.002). Cell viability following prolonged exposure to Chk1i also demonstrated a similar trend of increasing sensitivity for UWB1 (25.3 ± 16%) followed by UWB1-R (56.2 ± 17%) and then UWB1-WT (74.2 ± 34%). When used in combination with olaparib, both the ATRi and Chk1i significantly reduced cell viability for the UWB1 (0% and 0% cell viability respectively) and UWB1-R (0% and 1.4 ± 1% respectively) cell lines and was significantly lower compared to UWB1-WT (67 ± 19% and 76 ± 16% respectively) for the Olap + ATRi (*p* = 0.0005) combination and Olap + Chk1i (*p* = <0.0001) combinations.

## 4. Discussion

In this study, we investigated rationale drug combinations to overcome PARPi resistance in *BRCA* mutant ovarian cancer [18]. We first developed a resistant cell line by exposing a *BRCA1* mutant OC cell line, UWB1, to a physiologically relevant concentration of olaparib-containing media (1 µM) for 21 days. The olaparib-resistant cell line, UWB1-R, was significantly more resistant to olaparib compared to UWB1 parent cell line.

BRCA1/2 proteins function in two distinct cellular processes; specifically, the execution of HRR and the protection of stalled RF. As such, proliferating *BRCA1* mutant ovarian cancer cells are typically sensitive to PARPi and treatment can result in a rapid lethal accumulation DNA DSB. Over time, however, clinical responses are often incomplete and tumor cells can develop an acquired PARPi resistance [26]. Although this study did not investigate specific mechanisms of resistance, there is increasing evidence that drug resistance such as acquired olaparib resistance may involve a heterogeneity of resistant mechanisms instead of a singular driver mechanism [13,27]. Given the prospect that acquired olaparib resistance in our UWB1-R cells developed by a heterogenous collection of resistance mechanisms; we hypothesized that *BRCA1* olaparib resistant cells (UWB1-R) would have an increased dependence on key upstream pathway proteins. Of the known mechanisms of PARPi resistance based on in vitro and in vivo data, the mechanism of restored of RF stability and reactivation of HRR in *BRCA1* deficient cells are identified as important resistant pathways in this study [11]. It is well recognized that both ATR and Chk1 proteins both play a key coordinator role for RF stability and DDR. Related, Kim et al., using *BRCA1* and *BRCA2* mutant OC cell lines, showed an increased activation of ATR and Chk1 proteins after PARPi treatment, suggestive of an increased dependence of *BRCA1/2* mutant cells on the ATR/Chk1 pathway with PARPi therapy [28]. Additionally, RAD51 plays an important role in these pathways, albeit further downstream. By inhibiting these proteins, we hypothesized we could impact one or both mechanisms of resistance, thus restoring PARPi sensitivity. We believe our results, demonstrating synergy for both combinations of PARPi + ATRi/Chk1i, are explained at least in part by inhibiting these known resistance mechanisms. We also hypothesized that inhibiting G2 cell cycle proteins such as Wee1 and Chk1 may also be an important strategy to overcome PARPi resistance; however, our work did not uncover significant synergy for a PARPi + Wee1i combination.

ATR and its downstream kinase Chk1 are critical coordinator proteins which are highly activated by DNA replication stress, DDR including the repair of DSBs, nucleotide excision repair and inter-strand crosslink repair [29,30]. Additionally, ATR/Chk1 are activated for RF protection and stabilization [14,31,32,33] and ATRi has been shown to previously to increase recruitment of nuclease MRE-11 and result in enhanced RF degradation [13,34,35]. Using cell proliferation assays to screen several key drug inhibitors including ATRi (VE-821) and Chk1i (MK-8776) we found a significant increased sensitivity of the UWB1-R (*p* = 0.04) and UWB1-WT cells but not to UWB1 cell to single agent ATRi treatment (Figure 2B). Others have reported that ATRi resulted in an increased fork degradation for olaparib resistant cells but not UWB1 cells [13]. A similar trend of increased sensitivity to Chk1i (though the difference was not statistically significant) for the UWB1-R and UWB1-WT cells was observed compared to the UWB1 cells (Figure 2B). Overall, the increased sensitivity of the UWB1-R cells compared to UWB1 cells to ATRi and a similar trend for Chk1i, suggests a role of increased reliance and on ATR/Chki pathway in our olaparib resistance cells.

ATRi has previously been shown to sensitize cells to PARPi in the presence of a *BRCA1* mutation [36]. Additionally, drug synergy has been observed for Olap + ATRi for both HR-proficient *and* HR-deficient cell types. Here we show, that Olap + ATRi is synergistic not only with *BRCA1* mutant cells (UWB1) but also with *BRCA1* mutant olaparib resistant cells (UWB1-R) (Figure 3A). The demonstrated synergy with the UWB1-R cells was significantly less compared to UWB1 cells, but greater than wild type cells (UWB1-WT). Our results demonstrate that olaparib and ATRi are a synergistic drug combination over a wide concentration range that is clinically achievable in both *BRCA1* mutant olaparib-resistant and -sensitive cell lines, and suggest that Olap + ATRi is an appealing treatment strategy for both *BRCA1* mutant olaparib naïve and olaparib resistant OC. An ongoing Phase 2 clinical trial is open (CAPRI Trial, NCT03462342) evaluating olaparib plus AZD6738, an ATRi, for patients with germline *BRCA1/2* mutant recurrent OC, with both PARPi naïve and resistant disease cohorts. Enrollment is anticipated to be completed by the end of 2020. Additionally, there is a Phase I trial scheduled (NCT04149145) to be opened in early 2020 for advanced serous PARPi resistant OC evaluating niraparib plus M4344, an ATRi.

The combination Olap + Chk1i is also an attractive anticancer strategy based on synergy observed in both UWB1 and UWB1-R (Figure 3A), likely due to Chk1′s known central role in G2 cell cycle control, DDR and RF stability. Further, Jelinic et al., have previously demonstrated that olaparib increases G2 phase arrest in treated tumor cells [37], thus supporting a potentially novel strategy of combining for Chk1i with a PARPi for the treatment of OC. These are the first results to our knowledge demonstrating a synergy for Olap + Chk1i with olaparib resistant cells. Collectively our results are encouraging that Olap + Chk1i shows synergistic activity towards both *BRCA1* mutant OC cancer cells which either olaparib naïve or olaparib resistant. Currently there are no clinical trials we are aware of combining a PARPi + Chk1i for *BRCA* mutant OC.

Having established from the proliferation assays that Olap + ATRi and Olap + Chk1i treatment combinations are synergistic in UWB1-R cell lines, we sought to determine if either treatment strategy was effective with long term drug exposure and to what extent acquired olaparib resistance can be overcome for each combination. From the long-term drug exposure study results (Figure 4), olaparib monotherapy did not eliminate cell growth for either the olaparib-naïve *BRCA1* cells or olaparib resistant *BRCA1* cells. The effect of adding either ATRi or Chk1i to olaparib was highly effective at eliminating both UWB1 and UWB1-R cell populations. These combinations were likewise significantly less active towards UWB1-WT cells. It is worth noting, our results demonstrating drug synergy for the Olap + ATRi and Olap + Chk1i combinations (Figure 3,4) are in line with earlier results from Kim et al., for other *BRCA1/2* mutant, olaparib naïve, OC lines [28]. Additionally, our study demonstrates that both drug combinations are also highly effective as a cytotoxic treatment to *BRCA1* mutant, PARP-resistant OC cells. Lastly, our results suggest that cytotoxic responses for the combination treatments could be stratified by *BRCA1* status (Figure 5). Although only 10–20% of ovarian cancers have germline *BRCA1/2* mutations [2,38,39,40], up to 51% of high-grade serous OC tumor cells contain either a genetic or epigenetic inactivation of *BRCA*1/2 [16], making both treatment combinations an attractive strategy for recurrent *BRCA1/2* mutant OC.

While our results demonstrate Olap + ATRi/Chk1i is an effective treatment strategy in our experimental system, there are several limitations to our work. First, the current study focused on a *BRCA1* germline mutation and although *BRCA1* and *BRCA2* both have roles in HRR and RF stability, their specific molecular functions in each pathway are distinct and it is unclear if similar results would be extended to *BRCA2*-deficient OC cells. Additionally, olaparib was chosen as the PARPi for our experiments. Although we did screen a second PARPi, niraparib, and found an increased resistance of the UWB1-R cells to niraparib as well (data not shown), there is evidence that the binding affinity [41] and off-target effects of different PARPi may differ [37]. Lastly, translational experiments from in vitro to in vivo should be considered using a PDX or an equivalent biologically relevant model system with future work. Despite the aforementioned limitations, our study speaks to a clinically important issue in the treatment of OC, namely, treatment strategies towards overcoming PARPi resistance in *BRCA1/2* mutant OC. We find that for the treatment approaches used here, the outcomes for high grade serous OC may be stratified by BRCA status, and in the case of *BRCA1* mutant OC, an ATRi or Chk1i may be combined with olaparib to overcome acquired olaparib resistance.

## Figures and Tables

**Figure 1 diagnostics-10-00121-f001:**
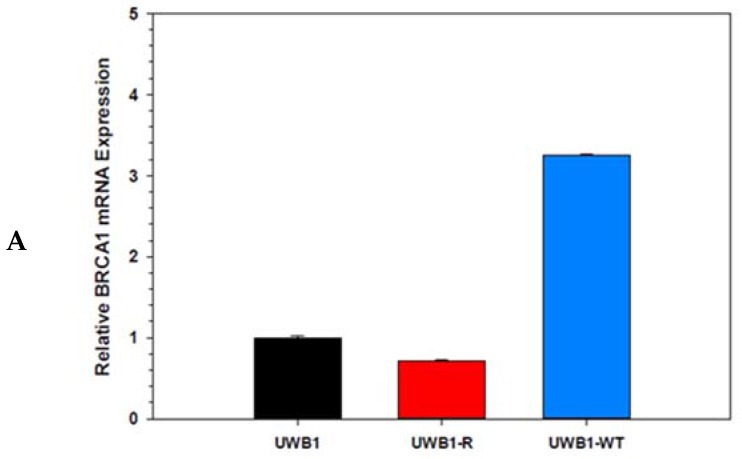
(**A**) Total RNA was extracted from UWB1, UWB1-R and UWB1-WT cell lines. mRNA expression of BRCA1 is shown relative to UWB1 cell line. Quantification of RNA levels relative to UWB1 were calculated by the Livak method using the QuantStudio PCR software analysis. Experiments run in triplication with standard deviation error bars shown. (**B**) Cell viability assay shown for olaparib-treated cells. Cells were treated in olaparib-containing media for 144 h at increasing doses and the dose-response plotted to estimate the IC_50_. Experiments were run in triplicate with standard deviation error bars shown. (**C**) Estimated IC_50_ comparing UWB1 (1.6 ± 0.9 µM) versus UWB1-R (3.4 ± 0.6 µM) and was statistically significant (α < 0.05). UWB1-WT cells were highly resistant to olaparib treatment with an IC_50_ was not attained in our (>10 µM). Plotted are the individual means with standard deviation error bars shown. α ≤ 0.05 = *.

**Figure 2 diagnostics-10-00121-f002:**
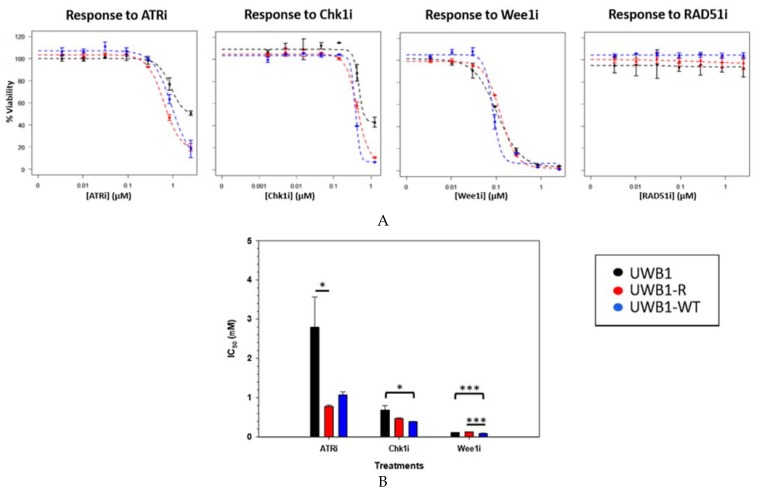
(**A**) Cell viability assay shown for cells versus screened inhibitors: ATRi, Chk1i, Wee1i and RAD51i. Cells were treated with media containing inhibitors for 144 h at increasing doses and the dose-response plotted for each cell line to estimate the IC_50_. Experiments were completed in duplicate with standard deviation error bars shown. (**B**) Estimated IC_50_ for each cell line. IC_50_ of ATRi was estimated to be 2.8 ± 0.77 µM, 0.78 ± 0.04 µM and 1.1 ± 0.08 µM for the UWB1, UWB1-R and UWB1-WT cells, respectively. IC_50_ of UWB1-R cells was significantly lower compared to UWB1 cells (α < 0.05). IC_50_ for the Chk1i was: 0.69 ± 0.11 µM, 0.47 ± 0.02 µM and 0.39 ± 0.004 µM respectively. UWB1-WT was significantly lower than UWB1 (α < 0.05). IC_50_ for the Wee1i: 0.11 ± 0.005 µM, 0.13 ± 0.05 µM, and 0.09 ± 0.005 µM for UWB1, UWB1-R and UWB1-WT respectively. UWB1-WT was significantly lower than UWB1 and UWB1-R cells (α < 0.001).

**Figure 3 diagnostics-10-00121-f003:**
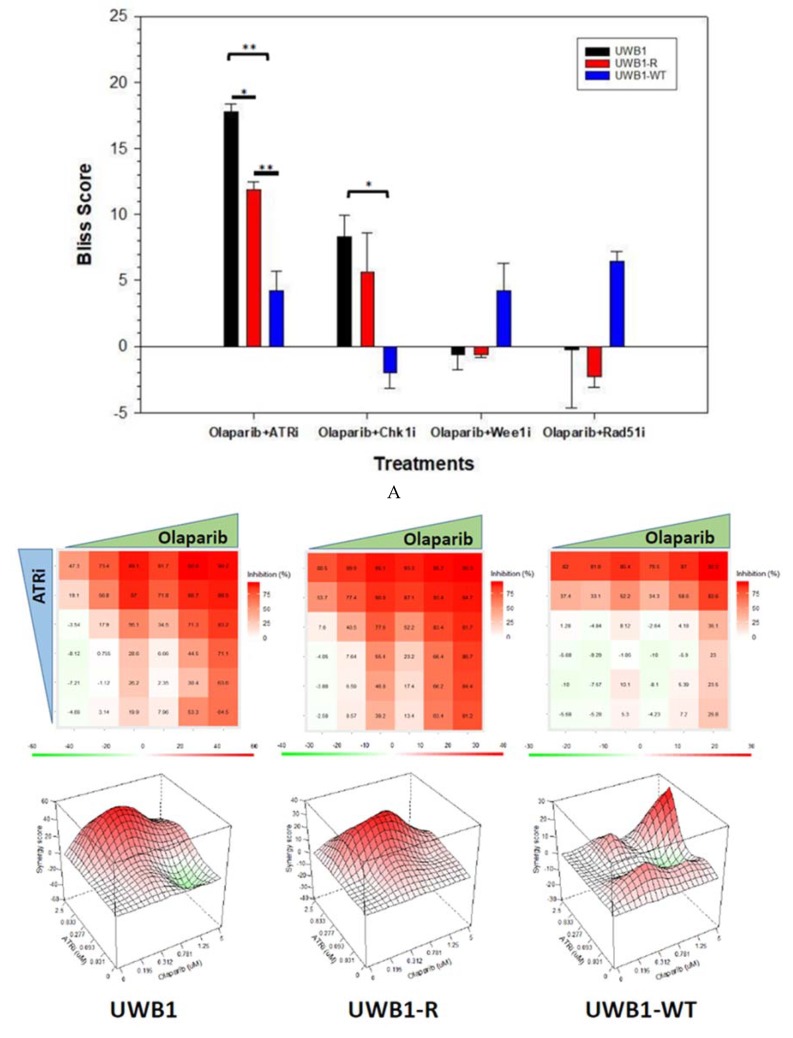
(**A**) Overall estimated Bliss scores for UWB1, UWB1-R and UWB-WT cells for each drug combination of olaparib with: ATRi (Olap + ATRi), Chk1i (Olap + Chk1i), Wee1i (Olap + Wee1i), RAD51i (Olap + RAD51i). Olap + ATRi resulted in an overall synergy response for all cell lines with UWB1-WT significantly less than UWB1 and UWB1-R. Olap + Chk1i also resulted in an overall synergy response for UWB1 and UWB1-R cells, but antagonism for UWB1-WT cells. Olap + Wee1i and Olap + RAD51i combinations did not result in any significant synergy for the UWB1 and UWB1-R cells, but synergy was observed for the UWB1-WT cells. α < 0.05 = *, α < 0.01 = **, α < 0.001 = ***. Experiments were completed in duplicate with standard deviation error bars shown. (**B**) 2-D surface response for cell inhibition and 3-D surface Bliss synergy response for Olap + ATRi for UWB1, UWB1-R and UWB-WT cells. (**C**) 2-D surface response for cell inhibition and 3-D surface Bliss synergy response for Olap + Chk1i for UWB1, UWB1-R and UWB-WT cells.

**Figure 4 diagnostics-10-00121-f004:**
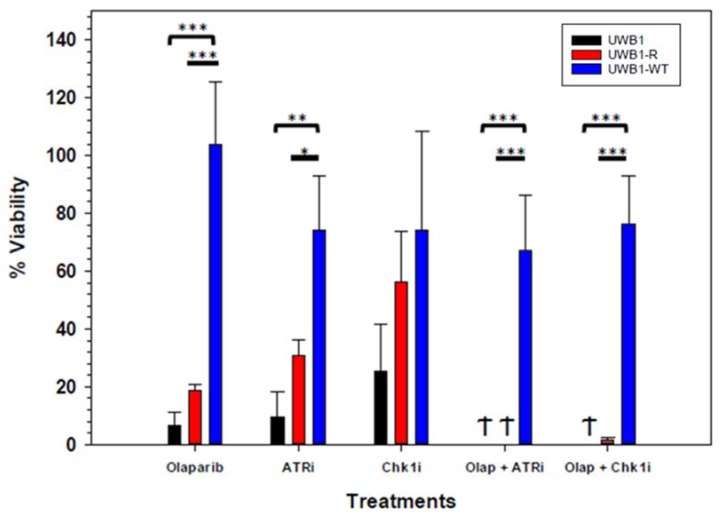
Cell viability after 21 days exposure to olaparib (1 µM), ATRi (0.8 µM), Chk1i (0.5 µM), and combination treatments with olaparib (1 µM) + ATRi (0.8 µM) and olaparib (1 µM) + Chk1i (0.5 µM). Olaparib treatment: UWB1 and UW1-R cells were significantly more sensitive to olaparib and ATRi (*p* = 0.002). ATR treatment: Cell viability for UWB1-R (*p* = 0.002) and UWB1-WT cells (*p* = 0.02) were significantly less compared to UWB1 cells. Chk1 treatment: Increasing cell viability trend is noted for UWB1-R compared to UWB1 followed by UWB1-WT. The magnitude of the differences was not statistically significant. Olap + ATRi treatment: Cell viability was significantly less in UWB1 (0%) and UWB1-R cells (0%) compared to UWB1-WT (67 ± 19%) (*p* = 0.0005). Olap + Chk1 treatment: Cell viability for UWB1 (0%) and UWB1-R (1.4 ± 1%) was was significantly less compared to UWB1-WT (76.4 ± 16%) (*p* = <0.0001). Experiments were completed in triplicate with standard deviation error bars shown. α < 0.05 = *, α < 0.01 = **, α < 0.001 = ***.

**Figure 5 diagnostics-10-00121-f005:**
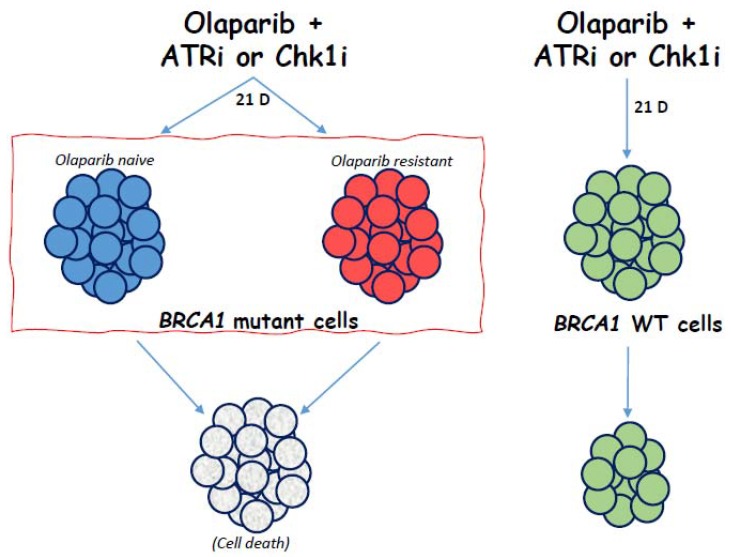
Treatment of *BRCA1* mutant cells, regardless of olaparib-resistant status, results in cell population death with Olaparib + ATRi or Olaparib + Chk1i. *BRCA1* wild-type cell population is decreased only slightly with the same treatment.

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
