# Peer review of "Olaparib Combined with an ATR or Chk1 Inhibitor as a Treatment Strategy for Acquired Olaparib-Resistant BRCA1 Mutant Ovarian Cells"

_diagnostics, 2020, doi:10.3390/diagnostics10020121_

Round 1
Reviewer 1 Report
In this manuscript, the role of Olaparib Combined with an ATR or Chk1 Inhibitor as a Treatment Strategy for Acquired Olaparib Resistant BRCA1 Mutant Ovarian Cells was studied. This is an interesting study in an area that needs to be researched (i.e. PARPi resistance). However, there are a few issues that need to be resolved before publication.
Abstract needs rewriting in the results section. Not everybody understands what a Bliss score is. Under methodology better description of Bliss independent model is required. Figure 1A and 1C: poor labelling of Y-axis for both figures Figure 1A: Western blotting and/or sequencing would have been more convinving Figure 2A: poor labelling description Under discussion, the authors should critically discuss known pathways implicated in PARP resistance.
Reviewer 2 Report
The study conducted by Burgess et al describes the effects on cell proliferation of PARPi (Olaparib) as monotherapy and in combination with ATR, Chk1, Wee1 and RAD51 inhibitors using PARPi sensitive and resistant BRCA1 mutated ovarian cancer cell lines as models. The authors demonstrated that ATRi and Chk1i have a synergistic effect with Olaparib decresing tumor cell growth. The study highlights the need to introduce new therapies in combination with PARPi for the treatment of BRCA1 mutated patients that frequently develop resistance.
Major point:
The manuscript is partially innovative because similar study was conducted by Kim H et al. These authors described that ATR/CHK1 blockade increased cell death and tumor regression in BRCA1/2MUT HGSOC cells (Clinical Cancer Research 2017) when use in combination with PARPi. The innovative results described in this work concern the resistant BRCA1 mutated ovarian cancer cell lines that does not described by elsewhere and the effects obtained using PARPi and Wee1 or RAD51 inhibitors.
Minor point:
The legends of Figure 1 A and C, 2B are missed
Author Response
Response to Reviewer 2 Comments
1. The manuscript is partially innovative because a similar study was conducted by Kim H et al.
These authors described that ATR/CHK1 blockade increased cell death and tumor regression in
BRCA1/2 mutant HGSOC cells (Clinical Cancer Research 2017) when used in combination with
PARPi. The innovative results described in this work concern the resistant BRCA1 mutated
ovarian cancer cell lines that is not described elsewhere and the effects obtained using PARPi
and Wee1 or RAD51 inhibitors.
Thank you for your feedback.
2. The legends of Figure 1A and C, Figure 2B are missed.
Corrections have been made to the axis labeling for Figure 1 and 2.
Round 2
Reviewer 1 Report
I am happy with the edits.
Reviewer 2 Report
The discussion needs to be changed including a comment regarding the paper of Kim H et al previously cited.They should discuss the results in light of those reported by Kim H et al, emphasizing the innovation and the weakness of the paper.
Round 3
Reviewer 2 Report
The manuscript is now accettable for publication